# Utilization of Recycled Egg Carton Pulp for Nitrocellulose as an Accelerant in Briquette Production

**DOI:** 10.3390/polym15132866

**Published:** 2023-06-28

**Authors:** Amena Andok, Seng Hua Lee, Melissa Sharmah Gilbert Jesuet, Ismawati Palle

**Affiliations:** 1Faculty of Tropical Forestry, Universiti Malaysia Sabah, Jalan UMS, Kota Kinabalu 88400, Sabah, Malaysia; amenaandok18@gmail.com (A.A.A.); melissa.gilbert@ums.edu.my (M.S.G.J.); 2Department of Wood Industry, Faculty of Applied Sciences, Universiti Teknologi Mara (UiTM) Cawangan Pahang Kampus Jengka, Bandar Tun Razak 26400, Pahang, Malaysia; leesenghua@hotmail.com

**Keywords:** nitrocellulose, briquette, egg carton pulp, bleached, accelerant

## Abstract

Nitrocellulose (NC) is a conservative material that is used in a variety of applications, such as coating agents, biodegradable plastics, and propellant main charge. Nitrocellulose raw materials are easily obtained from lignocellulose sources, most notably cotton and wood pulp. The egg carton, a recycled paper waste designed for packaging and transporting eggs, is used in this study to make nitrocellulose in pulp form. The effects of different nitration durations (40, 50, and 60 min) from egg carton pulp bleached with various KOH concentrations (0.6 M, 1.0 M, and 1.5 M) on NC properties were evaluated. The accelerant properties of the NC of nitration time in 50 min were studied in a rice husk charcoal briquette. Rice husk charcoal briquettes are made in various ratios with nitrocellulose as an accelerant (97:3, 96:4, and 95:5). The NC was characterized using Fourier transform infrared (IR) spectroscopy and thermogravimetric (TG) analysis. 1.0 M of bleached egg carton pulp has the highest cellulose content (86.94%) with the presence of crystalline structure of cellulose at peak 1430 cm^−1^ after the bleaching process. Meanwhile, different nitration times revealed that 50 min had the highest nitrogen content (7.97%) with a 1.23 degree of substitution (DS) value. Based on its TG analysis, NC 50 has met the requirements for use as an accelerant for briquettes, with an onset temperature of 91.60 °C and a weight loss of 62.60%. Infrared at peak 1640 cm^−1^ confirmed the presence of NO_2_ groups in nitrocellulose successfully formed by nitration. After the addition of nitrocellulose, the calorific value of the briquette increased from 13.54% to 15.47%. Fixed carbon and volatile matter showed the same pattern. The combustion of nitrocellulose-briquette has also been demonstrated by Td10% of degradation, which degraded between 310 and 345 °C.

## 1. Introduction

Nitrocellulose is a cellulosic substance that is formed by reacting ordinary cellulose with a mixture of nitric and sulfuric acid, resulting in the substitution of (–OH) groups for (–NO_2_) groups [1]. Nitrocellulose, also known as gun cotton, is one of the primary ingredients in smokeless gunpowder [2]. The most infamous use of nitrocellulose in the 1890s was as a film base, but it had drawbacks that led to a significant shift to its sole use in explosives [3]. When directly exposed to a hot and humid environment, this nitrocellulose easily self-ignites. The nitrogen content of nitrocellulose determines the application and use of this polymer. Nitrocellulose with a high nitrogen content (>12.5%) is typically used for explosives, whereas nitrocellulose with a low nitrogen content was discovered to be an excellent product for wood coating, nail lacquer, automotive paints, and leather finishes [4]. The nitrogen content of nitrocellulose has a significant impact on its thermal stability [5] because the decomposition temperature decreases with increasing nitrogen content, making it flammable. Aside from its flammability, nitrocellulose is soluble in most organic solvents, which allows it to be used as an alternative cellulose derivative in polypropylene products [6]. Nitrocellulose can be used to make wood coating, nail lacquer, automotive paints, and leather finishers with a low nitrogen content [4].

Purified cotton linters and wood pulp are the most common sources of cellulose used in the production of nitrocellulose. Muvhiiwa et al. [7] created nitrocellulose from tobacco stalks and used the soda pulping method to create cellulose pulp. *Acacia mangium* was also used as a raw material for the pulp production of nitrocellulose because it contains more than 80% cellulose. Aside from that, there are non-wood products for nitrocellulose production, such as Schimansky [8], who used cotton linters as the main cellulose source for the nitrocellulose product. However, no studies have been conducted to date on the preparation of NC from recyclables such as egg cartons. The egg carton is one of the most common fibrous wastes used for egg placement. The egg carton is primarily made of molded cellulosic pulp products containing 6–17% residual lignin [9]. The bleaching process is a promising method for improving cellulose by removing lignin and hemicellulose, which reduces the degree of polymerization [10]. Meanwhile, the alkaline treatment has been shown to be an effective method for exposing cellulose’s crystalline structure and improving cell wall accessibility [7]. Pretreatment with potassium hydroxide (KOH) as a strong alkali rather than sodium hydroxide can significantly increase cellulose accessibility (NaOH) [11].

One of the most common issues in the production of briquettes nowadays is their limited ignition [12]. As a result, introducing an accelerant for briquette formation could be one of the alternative methods for improving ignition. Rice husk charcoal is one of the most commonly used potential raw materials in the production of briquettes. Rice husk is a byproduct of the rice milling process that is easily collected. Rice husk is typically burned and left in the field without being reused. As a result, many researchers are concerned about using rice husk as a renewable and recyclable material in briquette production. The briquetting process is a treatment procedure that involves mashing, mixing raw materials, printing with a hydraulic system, and drying under specific conditions to produce briquettes with a form, physical dimension, and specific chemical properties. In order to produce adequate heat, good-quality briquettes must be in the proper shape. The primary challenge in producing briquettes is determining the proper composition so that the briquette’s heating value increases and its use increases [13].

Because egg cartons are recyclable, converting them to cellulose is a sustainable option. In this study, egg carton pulp was bleached with 0.6 M, 1.0 M, and 1.5 M KOH as a bleaching agent and then used to make nitrocellulose. Nitrocellulose was produced from egg carton pulp using a nitration process with varying reaction times (40, 50, and 60 min). FTIR was used to analyze the nitrocellulose product, and TGA was used to determine the temperature of degradation. Various nitration times were used to produce nitrocellulose with varying nitrogen content, degree of substitution, and solubility. Furthermore, this research provides theoretical support for the use of egg carton pulp in the production of nitrocellulose. The goal of this study was to determine the optimal nitrocellulose-to-briquettes ratio in order to produce fuel briquettes with a high calorific value, volatile matter, and fixed carbon.

## 2. Materials and Methods

### 2.1. Materials

Common egg cartons were retrieved from local eateries and used as the main cellulosic material. It undergoes hydro-pulping to obtain individual strains of fiber before being sieved and screened into a uniform size of 30-mesh size using a sieve shaker. Potassium hydroxide (KOH), distilled water, ethanol, toluene, 72% sulfuric acid, sodium chloride, 10% acetic acid, acetone, sodium hydroxide, acetic acid, pure sulfuric acid, and nitric acid were reagent grade and were used without further purification.

### 2.2. Bleaching Process

The subsequent bleaching to further remove lignin content was conducted with potassium hydroxide (KOH) following [10] methods with some modification. About 10 g of recycled egg carton pulp was placed in 190 mL of bleaching solutions with concentrations of 0.6 M, 1.0 M, and 1.5 M of KOH, respectively. The egg carton pulps were soaked in solutions by using 250 mL in a beaker and heated at 60 °C in a water bath for 120 min. Afterward, the remaining egg carton pulp was separated using crucible glass (No. 1) and subsequently washed with distilled water before oven drying at 105 ± 3 °C for 48 h.

### 2.3. Nitration Process for Nitrocellulose Production

The bleached egg carton pulps were soaked in the solution mixture of nitric acid and sulfuric acid. The reaction of egg carton pulp with a 1:1 ratio of undiluted nitric acid to sulfuric acid was utilized. This procedure involves immersing 2 g of pulp in 94 mL of the acid mix [6] at different reactions time (40, 50, and 60 min). To prevent instability, the wet nitrated pulp was immediately submerged in 250 mL of hot water (80 °C) after the active nitrating process. The product of nitrocellulose was dried in a desiccator until constant weight was achieved.

### 2.4. Formation of Briquette

The rice husk charcoal and nitrocellulose were weighed around 88–90% and 3–5% on weight percentage and blended at different mixing ratios of 97:3, 96:4, and 95:5, respectively. The blends of the rice husk charcoal and nitrocellulose were mixed with the binder (starch gel) and CaCO_3_. The weight percentage of the binder and CaCO_3_ in the briquette samples was 4% and 3%, which were constant at different mixing ratios. The feedstock was then poured into the prepared mold after blending. The compaction of the feedstock was performed using a 20-ton hydraulic press machine. A hydraulic press was used to press the coal with pressure of 50 kg/cm^2^. Each briquette had a holding time of 120 s [14]. The briquettes were dried in an oven for 8 h at 105 °C, cooled, and packed into polyethylene plastics to avoid re-adsorption of water [12].

### 2.5. Analytical Methods

The chemical components of treated and untreated recycled egg carton pulp were determined using a standard method. The moisture content of the pulp was determined following TAPPI standard T 264 by drying a sample at 103 ± 5 °C for 24 h. The ethanol-toluene extraction was attained following TAPPI Standard T207 cm-97, using Soxhlet extraction equipment. The hot- and cold-water solubilities were measured according to TAPPI standard T207 os-752 04 cm-88. Klason lignin was determined based on [15] standard by hydrolysis with 72% and 3% sulfuric acid. The holocellulose content was measured following [16], and α-cellulose was determined with a 17.5% sodium hydroxide solution [17].

The CHN elemental analyzer (Vario Macro Cube) was used to evaluate the degree of substitution (DS) and nitrogen content (N%) in nitrocellulose. The degree of substitution (DS) of bleached egg carton pulp nitrocellulose was measured following Equation (1) [18].
(1)$$\mathrm{DS}=\frac{1.62\times \%N}{14-0.45\times \%N}$$
where %N is the percentage of nitrogen content that obtained from elemental analyzer.

To investigate the influence of nitrating time on solubility, the dried NC (0.2 g) were dissolved in 10 mL of acetone for 1 h to achieve dissolution. The undissolved nitrocellulose was filtered off and left to dry [19]. The solubility of nitrocellulose in ethanol was characterized through Equation (2) [20]:(2)$$S=\frac{m-\left(m_{1}-m_{2}\right)}{m}\times 100$$
where m is the mass of nitrocellulose, m_1_ is the funnel mass with the dry residues, and m_2_ is the mass of the empty funnel.

The structure of egg carton pulp and nitrocellulose pulps were identified using Fourier transform infrared (FT-IR) spectroscopy in the range of 4000–400 cm^−1^ with Perkin Elmer Spectrum 400 FT-IR/NIR with Imaging System.

Thermogravimetric analysis (TGA) was performed using Mettler Toledo TGA SDTA 851e with a continuous flow of N_2_ atmosphere to investigate the change in properties of egg carton pulps and nitrocellulose as a function of temperature. The pulps (6–10 mg) were analyzed under an N_2_ atmosphere (20 mL min^−1^), with a heating rate of 10 °C min^−1^ with a temperature from 30 to 600 °C.

The calorific value of a briquette was determined by measuring the heat generated by the full combustion of a specified amount of it, expressed in calories per gram. A bomb calorimeter (AC500 Calorific Value BS EN 14918) was used to measure the calorific value. This test was carried out to determine the standard quality of the briquettes’ fuel power. Meanwhile, the percentage of fixed carbon was measured by Equation (3).
F_c_ (wt%) = 100 − (V_m_ + M_c_ + Ac)(3)
where F_c_ is the (wt%) fixed carbon obtained for each briquette sample, V_m_ is the (wt%) volatile matter, M_c_ is the moisture obtained, and A_c_ is the (wt%) ash content obtained for each briquette sample. The volatile matter is an unstable substance that frequently transitions quickly to another form or vaporizes, expressed as a percentage.

### 2.6. Statistical Analysis

Properties measurement data (ANOVA) was used to compare chosen parameters, and significant differences were identified with a *p* ≤ 0.05 in the comparison test. These analyses were performed using Statistical Analysis System (SAS).

## 3. Results and Discussion

### 3.1. Bleaching Process of Egg Carton Pulp

#### 3.1.1. Chemical Constituents of Bleached Egg Carton Pulp

Table 1 shows the chemical contents of bleached and unbleached egg carton pulp at various bleaching concentrations. The lignin content clearly decreased after the bleaching process. This is because the goal of alkaline delignification is to remove lignin in order to increase cellulose purity [21]. The primary raw material used in the production of nitrocellulose is pure cellulose [22]. The results show that 1.0 M has the least amount of lignin (1.01%) but the most cellulose (86.94%). The 1.5 M has more lignin than the 1.0 M, which could be due to the bleaching process reaching the final delignification phase. Gellerstedt [23] claims that the rate of lignin dissolution slows significantly with prolonged cooking. This could be because the extent of concentration affects the bleaching response of the pulp, making it increasingly difficult to remove the lignin [24].

Based on statistical analysis, there are significant differences in cellulose concentrations between 1.0 M and 1.5 M at *p* ≤ 0.05. The increased cellulose content could be attributed to the partial removal of hemicellulose and lignin during bleaching. Some lignin in fibers cannot be easily degraded due to the presence of strong carbon-carbon linkages that are resistant to chemical attack [25]. It is also thought that the hemicellulose residues may remain in the structure due to the stable hydrogen bonding of hemicellulose to cellulose fibrils.

The bleached egg carton pulp with the highest ethanol-toluene solubility (5.71%) and hot-water-soluble extractives was found at 1.0 M. (8.15%). The statistical analysis reveals that 1.0 M of ethanol-toluene solubility and hot water solubility differ significantly (*p* ≤ 0.05) from other concentrations. However, at 1.0 M, bleached egg carton pulp has the lowest cold-water solubility (5.96%), with no significant differences at 1.5 M. The existence of inorganic matter, tannins, gums, and sugar can be determined using cold and hot water solubilities [26]. Cold, hot, and ethanol-toluene solubility can all help with lignin content. The lower cold and hot water soluble contained more lignin and hemicellulose, according to Nazri et al. [27]. It demonstrates that some extractives react easily with pulping chemicals, while others are insoluble in water and can withstand the pulping process. This could be due to changes in the solubility of bleached egg carton pulp caused by pulp degradation, particularly holocellulose, which has a higher hydroxyl group [28].

After bleaching, the pH of egg carton pulp becomes more alkaline. Egg cartons are typically made of a combination of paper, water, and grass fibers, with the grass fibers providing the acidic content. According to De Veth and Kolver [29], the pH of ryegrass ranges between 5.8 and 6.6.

#### 3.1.2. Fourier Transform Infrared (FT-IR) Spectra Analysis of Bleached Egg Carton Pulp

Figure 1 shows the FT-IR spectrum for the unbleached and bleached egg carton pulp. The broad peak at 3330 cm^−1^ is attributed to the stretching vibration of the hydroxyl group [30]. This is due to the carbohydrates and vibrations of the hydroxyl group. The sensitivity of the band at 1640 cm^−1^ decreased with increasing molarity of KOH, which was interpreted as the loss in the C=C groups linking the aromatic skeleton in lignin [31]. A comparison of unbleached and bleached egg carton pulp shows the alkaline treatment disrupts the lignin structure and breaks the link between lignin and other carbohydrate components in egg carton pulp. The bands at 1430 cm^−1^ and 1030 cm^−1^ correspond to the –CH_2_ and C–O–C pyranose ring skeletal vibrations of cellulose, respectively, and it is related to the amount of the crystalline structure of the cellulose [32], where the peak intensity exhibited changes at 1430 cm^−1^. It is probably due to a bleaching process that could lead to the oxidation of cellulose [33]. The peak at 1370 cm^−1^ was attributed to the C–H bending bands in CH_3_ in the pulp. The decrease in relative intensity of the C–O band at 1160 cm^−1^ revealed that lignin or hemicelluloses were reduced during the bleaching process [34].

#### 3.1.3. Thermalgravimetric Analysis (TGA) of Bleached Egg Carton Pulp

The weight loss, decomposition temperature, and thermal stability of the bleached egg carton pulp were all evaluated using thermogravimetric analysis (Figure 2). Table 2 summarizes the thermal properties of the pulp, including the initial decomposition temperature (T_onset_), maximum degradation temperature (T_max_), and char residue. According to the onset temperature, 1.0 M bleached egg carton pulp has the highest temperature at 294.01 °C. The initial degradation increased from 0.6 M to 1.0 M before decreasing slightly when 1.5 M was reached. A similar pattern was observed for maximum degradation temperature (T_max_) values. The highest egg carton pulp degradation temperature indicates an increase in pulp crystallinity during the bleaching process [35]. Increased crystallinity increases heat resistance, which improves thermal stability. This is might due to the high lignin and extractives content in pulp (Table 1). According to Abdul Khalil [35], the higher amount of residue in pulp is directly proportional to the content of lignin, hemicelluloses, and extractive non-cellulosic components in raw fibers.

The degradation in the range of 198.9–200.1 °C was attributed to lignocellulosic component degradation, primarily hemicellulose, and some that remained unbleached [36]. Hemicellulose is easily decomposed due to its random and amorphous structure [37], which is especially easy to remove and degrade at low temperatures ranging from 220 to 315 °C. Unlike hemicellulose, cellulose has a long polymer of glucose, a good structure, is highly robust, and has high thermal stability. Several studies have found that bleaching improves thermal stability by gradually removing non-cellulosic material such as hemicellulose and lignin [38].

### 3.2. Characterization of Nitrocellulose from Bleached Egg Carton Pulp

#### 3.2.1. Degree of Substitution and Nitrogen Content of Nitrocellulose

Figure 3 depicts the degree of substitution (DS) of egg carton pulp nitrocellulose. Obviously, the DS of pulp samples increased as time passed. When compared to the others, the NC 50 has the highest DS value of 1.23%. Higher DS indicates more hydroxyl substitution in the monomeric unit by the reagent’s nitro groups. The highest theoretical DS value is three, implying that all three –OH groups can be replaced by nitro groups to produce a theoretically high nitrogen content (14.1%) [6]. Despite this, the number of hydrogen bonds cannot be appropriately adjusted to the number of –OH groups. This is because the number of hydroxyl oxygen atoms influences both the number of donors and acceptors [39]. However, after a certain point, the nitrocellulose begins to deteriorate as the reaction slows slightly. This could be due to the reverse reaction of nitrocellulose hydrolysis, which occurs more dominantly from the cellulose nitration reaction [40]. Increased nitration time accelerates the reverse reaction.

The nitrogen content of nitrocellulose follows the same pattern (Figure 3). The nitrogen content of nitrocellulose pulps increases with reaction time but decreases slightly after 60 min. DS and nitrogen content clearly have a direct relationship. The NC 50 (7.97%) has the highest nitrogen content, while the NC 40 has the lowest nitrogen content (7.04%). The nitrogen content of nitrocellulose is affected by the nitration time. Furthermore, because cellulose has many twists and turns, diffusion of the nitrating chemicals is limited, and nitration requires a reaction time [7]. According to statistical analysis, there are no significant differences in the degree of substitution and nitrogen content of nitrocellulose at *p* ≤ 0.05.

#### 3.2.2. Solubility of Nitrocellulose

Figure 4 depicts the effects of different nitration times on nitrocellulose solubility. Nitrocellulose with a nitrogen content greater than 6% is formed as the nitration period is extended. The higher the nitrogen content, the lower the solubility of the produced NC. Because of the high nitrogen content, it is typically dissolved in acetone, ethyl acetate, or ether-alcohol [41]. This also implies that the solubility of nitrocellulose produced from egg carton pulp is strongly dependent on the pulp’s nitrogen content. It influences the nitrocellulose structure, polymer solubility, and viscosity. Different nitrocellulose pulps have different nitrogen contents, as well as different solubility and applications [19].

#### 3.2.3. Thermal Degradation via Thermogravimetric Analysis (TGA) of Nitrocellulose

Figure 5 depicts the TGA and DTG analysis of egg carton pulp and nitrocellulose. All thermograms for all samples clearly show that the thermal decompositions occurred in a single stage. Table 3 depicts the maximum weight loss rate temperature of each nitrocellulose. It demonstrates that NC 50 has the lowest thermal degradation with the lowest weight loss rate temperature at 98 °C, indicating that it is suitable for combustion. Based on the onset temperature (Table 3), NC 50 degrades slightly slower than NC 40 but has a higher percentage of weight loss. The NC 50 degrades at 91.60 °C, with a percentage weight loss of approximately 62.60%. Meanwhile, based on a 10% weight loss (Td10%), NC 50 degraded faster than NC 60, as shown in Figure 5b. The weight loss occurs abruptly at various temperature ranges, which shows this temperature range is extremely narrow for decomposing of nitrocellulose under heat [42]. The thermal stability of nitrocellulose decreases as its nitrogen content increases (Figure 5). This could be attributed to its low thermal stability as a result of its flammable properties, which would make it more unstable when exposed to high temperatures [43]. This is due to the evaporation of remaining solvents and, in particular, the easily volatile products of membrane thermal degradation [44].

#### 3.2.4. Fourier Transform Infrared (FTIR) Spectra Analysis of Nitrocellulose

The FTIR spectra of egg carton pulp and nitrocellulose are shown in Figure 6. It reveals an intermolecular hydrogen bonding peak around 3350 cm^−1^. This band corresponds to residual hydroxyl groups of nitrocellulose. The peak appeared at 2880 cm^−1^ due to C–H stretching vibration. It shows that the band assignment is due to alkane CH stretching absorptions from the carbons of the glucose structures [45]. The absorption of 1717 cm^−1^ is the appearance of carbonyl groups (C=O) of nitrocellulose. It shows the intensity of C=O increases with the increasing nitration time. This band indicates an intermolecular interaction involving the nitrocellulose hydroxyl group (i.e., –C=O···H–O–) [46]. It shows that the spectrum of nitrocellulose is free from absorption in the region 1700–1800 cm^−1^, and it is attributed to the stretching of CN [47]. The carbonyl band shifted to lower wavenumbers (1717 cm^−1^) as a result of hydrogen bonding and did not overlap with the nitrate asymmetric stretching vibration (^v^_a_NO_2_) at 1640 cm^−1^ [34]. Nitrocellulose began to degrade with the breakdown of the O–NO_2_ link, producing NO_2_ gas at 1640 cm^−1^ [48]. The summaries of the IR spectrum of egg carton pulp and nitrocellulose are shown in Table 4.

### 3.3. Proximate Analysis of Briquette

Rice husk charcoal briquettes with varying ratios of rice husk charcoal to nitrocellulose content were created. In this study, three different ratios of rice husk to nitrocellulose (97:3, 96:4, and 95:5) were used to evaluate nitrocellulose properties, and the samples were labeled B2, B3, and B4, respectively. In contrast, briquettes with no nitrocellulose content were used as controls and designated as B1. Table 5 shows the calorific value of various nitrocellulose charcoal briquette ratios. The B2 briquettes had the highest calorific value at 15.47 MJ/kg, while the B3 briquettes had a lower amount at 14.88 MJ/kg before increasing slightly at B4 to 15.08 MJ/kg. The slight decrease in calorific value could be attributed to biomass blending and the addition of nitrocellulose. According to Sutrisno et al. [49], the moisture content of the briquette had a significant influence on its calorific value. Lower moisture content has a higher calorific value. Meanwhile, Tokan et al. [50] found that the calorific value of rice husk with varying moisture content ranged from 16.07 to 17.58 MJ/kg. Carbon, hydrogen, and oxygen are among the elements found in volatile matter. Based on Table 3, the briquettes were evaluated in terms of variable and fixed carbon and volatile matter. Nitrocellulose will increase the fixed carbon content as well as the volatile matter. Because the fixed carbon was a solid material containing ash, the weight percentage increased in proportion to the amount of volatile matter lost. This is due to an increase in volatile matter weight caused by the open pores of the rice husk charcoal expanding, which contributes to volatile matter loss [51]. The results show that increasing the nitrocellulose content resulted in an increase in the volatile matter percentage of the briquettes. Fuel samples with a high volatile matter content are ready to ignite. The high volatile matter in the briquette indicated that it would readily ignite with a proportionally large flame during combustion [52].

### 3.4. Thermogravimetric Analysis of Briquette

Figure 7 depicts a thermogravimetric analysis graph for briquettes based on the temperature of thermal degradation. Briquette weight loss was minimal at temperatures ranging from 50 to 200 degrees Celsius, corresponding to moisture removal [53], and maximum weight loss occurred at temperatures ranging from 300 to 400 degrees Celsius for all briquettes. Figure 7 depicts the Td10 percent of briquettes, with B3 having the lowest temperature of degradation (310 °C), followed by B4 (328 °C) and B2 (345 °C). Meanwhile, the B1 with 100 percent rice husk charcoal has the highest Td10% temperature of degradation. The Td10% of the briquette decreases as the nitrocellulose content increases. This could be because the nitrocellulose aided in the ignition of the briquette. The accelerant aided the charcoal briquette decomposition reaction to some extent. This causes an oxidative and transformed reaction of difficult-to-oxidize matter, as well as an increase in the release of coal volatile matter [54].

## 4. Conclusions

Because it is recyclable paper and easy to obtain, egg carton pulp was used as a raw material for accelerant in the production of briquette, and the additional accelerant in briquette was added to improve briquette ignition. This is due to the fact that briquettes take longer to heat up. Furthermore, the presence of cellulose content is an important source of nitrocellulose production. As the concentration of KOH increased, the amount of cellulose in the bleached egg carton pulp increased. The 1.0 M bleaching pulp had a high cellulose content of 86.94% and a low lignin content (1.01%). The presence of cellulose is indicated by the presence of a band at 1420 cm^−1^. Nitrocellulose is produced by nitrating cellulose, so the sources of cellulose are important. Increasing the concentration of KOH to 1.0 M resulted in high thermal degradation at Td10%, which could be attributed to the high percentage of cellulose. When the reaction time for the nitration process of egg carton pulp samples for nitrocellulose production was changed to 60 min, the nitrogen content and degree of substitution increased before decreasing. This is due to the reverse reaction, in which nitrocellulose takes precedence over the nitration reaction of cellulose due to hydrolysis. At the optimal reaction time of 50 min, nitrocellulose with the highest nitrogen content of 7.97% and 1.23 DS was obtained. Nitrocellulose dissolved in acetone with a high nitrogen content and variable solubility could be used in a variety of applications, including printing inks, nail lacquer, leather finishes, as well as gun cotton and flash papers. This was further demonstrated by thermal degradation, in which the NC 50 degraded faster and lost more weight than the NC 40. It demonstrates that nitrocellulose has lower thermal stability due to its combustible properties. Nitro groups were detected in nitrocellulose FTIR spectra that were successfully attached to egg carton pulp. It is expected that egg carton pulp will be a better candidate for NC resource and utilization. Based on briquette proximate analysis, the nitrocellulose content influences the calorific value, volatile matter, and fixed carbon content. The nitrocellulose content increased the energy content of the briquette, which is of high quality.

## Figures and Tables

**Figure 1 polymers-15-02866-f001:**
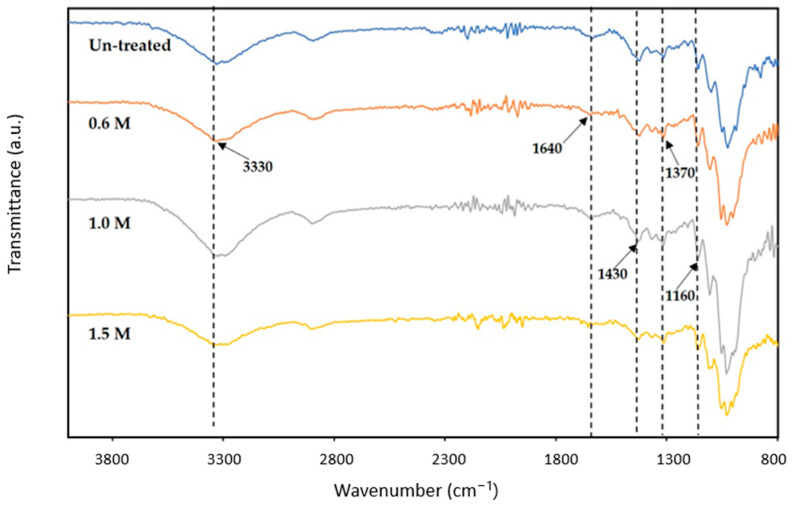
FT-IR spectra of bleached egg carton pulp.

**Figure 2 polymers-15-02866-f002:**
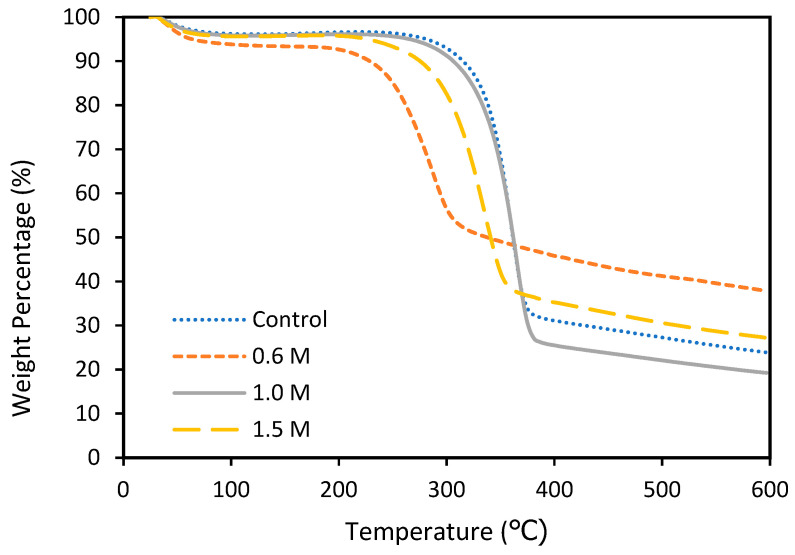
TG curve egg carton pulp at different bleaching concentrations at a heating rate of 10 °C min^−1^ of N_2_ atmosphere.

**Figure 3 polymers-15-02866-f003:**
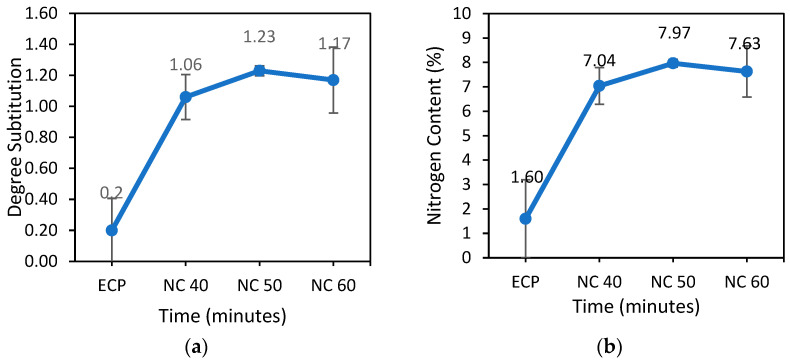
(**a**) Degree substitution and (**b**) nitrogen content of egg carton pulp and nitrocellulose.

**Figure 4 polymers-15-02866-f004:**
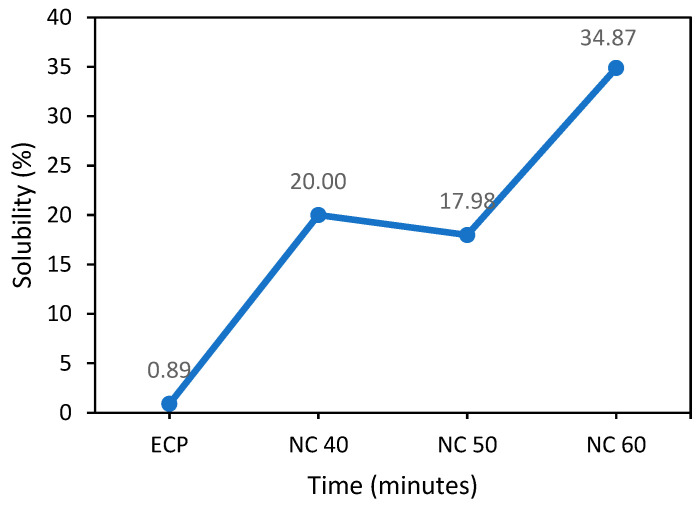
Solubility of nitrocellulose with different nitration times.

**Figure 5 polymers-15-02866-f005:**
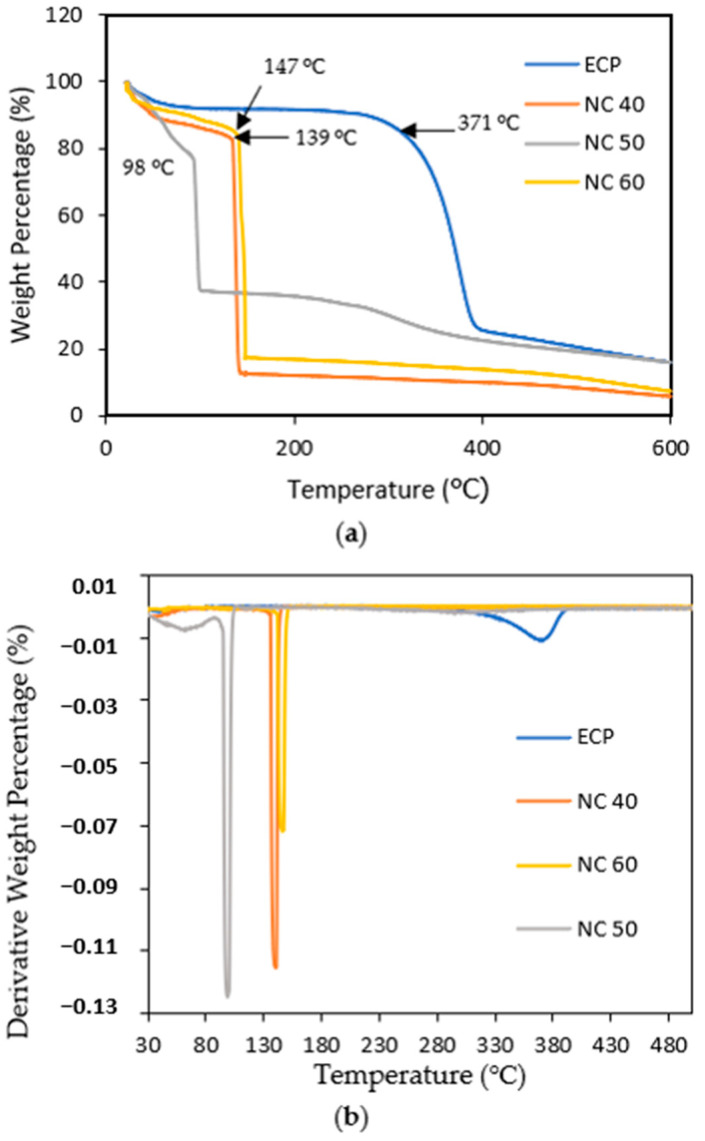
(**a**) TGA and (**b**) DTG curves of egg carton pulp and nitrocellulose.

**Figure 6 polymers-15-02866-f006:**
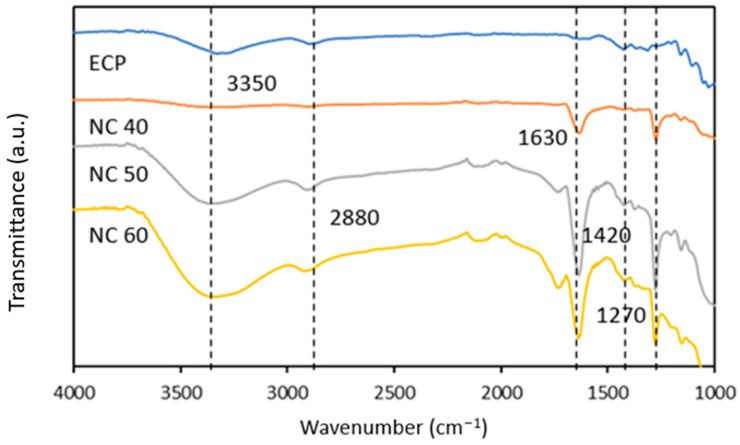
FTIR spectra of egg carton pulp and nitrocellulose.

**Figure 7 polymers-15-02866-f007:**
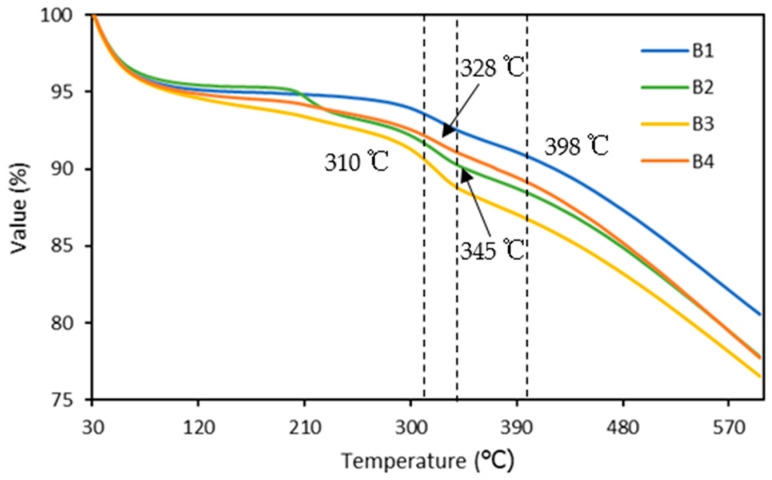
TGA curves of briquettes.

**Table 1 polymers-15-02866-t001:** Chemical constituents of egg carton pulp before and after bleaching.

Components	Control	0.6 M	1.0 M	1.5 M
Ethanol-toluene solubility	0.82 ^b^ (±0.23)	2.21 ^b^ (±0.89)	5.71 ^a^ (±0.42)	1.96 ^b^ (±0.97)
Cold water	3.67 ^a^ (±0.14)	11.36 ^c^ (±2.55)	5.96 ^ab^ (±0.25)	8.46 ^bc^ (±2.48)
Hot water	1.78 ^b^ (±0.13)	4.51 ^b^ (±1.48)	8.15 ^a^ (±2.93)	4.47 ^b^ (±1.82)
Klason–Lignin	6.63 ^a^ (±1.11)	4.03 ^b^ (±0.05)	1.01 ^c^ (±0.23)	2.68 ^d^ (±0.07)
Holocellulose	83.70 ^bc^ (±0.52)	81.40 ^c^ (±0.20)	87.89 ^a^ (±1.71)	85.86 ^ab^ (±1.68)
α-Cellulose	77.57 ^c^ (±1.25)	76.95 ^c^ (±0.43)	86.94 ^a^ (±1.96)	84.08 ^b^ (±1.90)
pH value	6.55 ^c^ (±0.26)	8.23 ^b^ (±0.22)	8.40 ^ab^ (±0.04)	8.67 ^a^ (±0.16)

Note: Means with the different letters a, b, c, and d in the same column were significantly different (*p* ≤ 0.05).

**Table 2 polymers-15-02866-t002:** Thermal properties of egg carton pulp at different bleaching concentrations.

Sample	T_onset_ (°C)	T_max_ (°C)	Residue (%)
Un-treated	294.54	352.40	23.85
0.6 M	210.78	277.11	37.83
1.0 M	294.01	354.30	19.21
1.5 M	259.68	325.94	27.18

**Table 3 polymers-15-02866-t003:** Thermal properties of egg carton pulp and nitrocellulose.

Samples	T_d10%_ (°C)	T_onset_ (°C)	Weight Loss (%)
ECP	243.26	336	8.24
NC 40	45.61	33.69	13.26
NC 50	51.06	91.60	62.60
NC 60	67.73	141.04	82.52

**Table 4 polymers-15-02866-t004:** Band assignment in IR spectra of bleached egg carton-nitrocellulose.

Frequency, cm^−1^	Assignment
**3350**	*v*(OH)(OH····OH) stretching [39]
**2880**	*v*(CH_2_) stretching
**1630**	σ(CH_2_) bending
**1420**	σ(COH) bending
**1270**	*v*_s_(NO_2_) symmetric valence
**1150**	*v*(C–O) stretching

**Table 5 polymers-15-02866-t005:** Proximate analysis of briquette.

Types of Briquette/Proximate Analysis	Calorific Value (MJ/kg)	Fixed Carbon (%)	Volatile Matter (%)
B1	13.54	23.28	25.64
B2	15.47	25.91	24.83
B3	14.88	28.07	26.95
B4	15.08	15.82	36.35

## Data Availability

The data presented in this study are available upon request from the corresponding author.

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
