# Peer review of "Utilization of Recycled Egg Carton Pulp for Nitrocellulose as an Accelerant in Briquette Production"

_polymers, 2023, doi:10.3390/polym15132866_

Round 1

Reviewer 1 Report

Dear Authors,

The paper is very well written and interesting, but there are some suggestions and comments that should be considered to improve the paper:

What is new about this study, other than that you used egg cartons for briquette production?

Line 22: Explain the abbreviation DS

Line 162: ….”The lignin content clearly decreased after the bleaching process Explain why the lignin content decreased and why this is important?

Table 1: There are missing explanations for the footnotes (a,b,c,d). Please explain them.

Table 2: Explain why the 0.6 M sample had the most residue after TGA analysis and why the 1 M sample had the least.

Line 378: “The B2 briquettes had the highest calorific value at 15.47 MJ/kg….” Explain why this is so!

Lines 377-391. The authors should compare the calorific value of the briquettes with the literature.

Line 412: “….could be used in a variety of applications”... for what applications?

Conclusion: What is the further use of briquettes from egg cartons? Explain in detail!

Author Response

Point 1: What is new about this study, other than that you used egg cartons for briquette production?

Response 1: The new about this study is the additional of accelerant in briquette. As we know that the most typical issues found when using charcoal briquettes made from biomass waste is their inability to ignite. So in order to enhanced its ignition, the nitrocellulose were used as an accelerant in briquettes.

Point 2: Line 22: Explain the abbreviation DS

Response 2: The abbreviation of DS has been explained as suggested

Point 3: Line 162: ….”The lignin content clearly decreased after the bleaching process” Explain why the lignin content decreased and why this is important?

Response 3: The bleaching process is used to dissolve the lignin and hemicellulose. This is because the higher amount of cellulose can be obtained by decreasing the amount of lignin. The effect of bleaching towards lignin has been explained as suggested

Point 4: Table 1: There are missing explanations for the footnotes (a,b,c,d). Please explain them.

Response 4: The letter “a, b, c and d” for the footnotes are specified as suggested

Point 5: Table 2: Explain why the 0.6 M sample had the most residue after TGA analysis and why the 1 M sample had the least.

Response 5: The 1.0 M shows the lowest amount of residue. This is might due to the high lignin and extractives content in pulp (Table 1). According to Abdul Khalil [50], the higher amount of residue in pulp is directly proportional to the content of lignin, hemicelluloses and extractive non cellulosic components

Point 6: Line 378: “The B2 briquettes had the highest calorific value at 15.47 MJ/kg….” Explain why this is so!

Response 6: The calorific value of briquette affects by moisture content and binder content. But, in this paper the binder content has fixed amount at 4%. So, this is might due to different value of moisture content gained in this process. The reason of the lower and higher calorific value has been added as suggested

Point 7: Lines 377-391. The authors should compare the calorific value of the briquettes with the literature.

Response 7: The other calorific value of the briquettes from other journals has been added as suggested

Point 8: Line 412: “….could be used in a variety of applications”...for what applications?

Response 8: “….could be used in a variety applications includes printing inks, nail lacquer, leather finishes, as well gun cotton and flash papers

Point 9: Conclusion: What is the further use of briquettes from egg cartons? Explain in detail!

Response 9: The egg carton pulp were act as an accelerant for charcoal briquette due to its slower to ignite. The purpose of additional accelerant in briquette was due to its takes longer time to burn up. The further uses of egg carton in briquette has been explained as suggested

Reviewer 2 Report

The main goal of the manuscript is to determine the optimal nitrocellulose-to-briquettes ratio to produce fuel briquettes with a high calorific value, volatile matter, and fixed carbon. So, it represents an important issue for renewable energy. However, before it is considered for publication, some improvements must be executed.

Abstract: The authors wrote, “Briquette production revealed that the calorific value, volatile matter, and fixed carbon contents was influenced by nitrocellulose content.” To show the effect of nitrocellulose on the briquette's properties more clearly, the authors could include the percentage of improvement after its addition. 

The abstract could contain information about TGA and FTIR.

The information about the rice husk charcoal must contain in the abstract.

The terms of Equation 2 must be specified.

Page 3, lines 137 – 140: the flow of the nitrogen atmosphere must be included in the text.

Page 3, lines 142 – 143: The specifications of the calorimeter must be included. 

In Table 1, the letters “a, b, c, and d” must be specified.

 In Figure 1, it must be included a legend to explain the spectrum.

In the section titled "3.1.3 Thermal Gravimetric Analysis (TGA) of Bleached Egg Carton Pulp” the word Thermal Gravimetric could be substituted for Thermogravimetric.

The title of Figure 2 could be more explanatory, including information about the thermal analysis parameters such as heating rate and atmosphere.

In Figure 5, it must be included DTG to improve the discussion of the degradation events.

Page 9, line 342: “The FTIR spectra of egg carton pulp and nitrocellulose are shown in Fig. 4.”. The authors must be substituted the number of Figure 4 for 6.

FTIR spectra of egg carton pulp and nitrocellulose could be improved, including the main absorptions. 

The call of Table 4 must appear in the text. 

Page 10, line 382: The authors must be substituted the number of Table 3 for 5. 

In Table 5, acronyms B1, B2, B3, and B4 must be specified.

The conclusion is consistent with the evidence and arguments.

The references are appropriate. 

Moderate editing of the English language is required. 

Author Response

Point 1: Abstract: The authors wrote, “Briquette production revealed that the calorific value, volatile matter, and fixed carbon contents was influenced by nitrocellulose content.” To show the effect of nitrocellulose on the briquette's properties more clearly, the authors could include the percentage of improvement after its addition.

Response 1: The percentage of improvement on briquette properties after additional of nitrocellulose has been included as suggested  

Point 2: The abstract could contain information about TGA and FTIR.

Response 2: The information of TGA and FTIR analysis of bleached egg carton pulp with different concentration of KOH, nitrocellulose with different nitration time and nitrocellulose-briquette production has  included as suggested

Point 3: The information about the rice husk charcoal must contain in the abstract.

Response 3: The production of rice husk charcoal briquette with nitrocellulose accelerant by different ratio are included as suggested

Point 4: The terms of Equation 2 must be specified.

Response 4: The terms of Equation 2 are specified as suggested

Point 5: Page 3, lines 137 – 140: the flow of the nitrogen atmosphere must be included in the text.

Response 5: The flow of the nitrogen atmosphere are included as suggested

Point 6: Page 3, lines 142 – 143: The specifications of the calorimeter must be included.

Response 6: The specifications of the calorimeter has included as suggested

Point 7: In Table 1, the letters “a, b, c, and d” must be specified.

Response 7: The letters “a, b, c and d” are specified as suggested

Point 8: In Figure 1, it must be included a legend to explain the spectrum.

Response 8: The legend of the spectrum in Figure 1 are included as suggested

Point 9: In the section titled "3.1.3 Thermal Gravimetric Analysis (TGA) of Bleached Egg Carton Pulp” the word Thermal Gravimetric could be substituted for Thermogravimetric.

Response 9: The thermal gravimetric analysis has been revised and changed to “Thermogravimetric” as suggested

Point 10: The title of Figure 2 could be more explanatory, including information about the thermal analysis parameters such as heating rate and atmosphere.

Response 10: The thermal analysis parameters such as heating rate and atmosphere are included in Figure 2 title as suggested

Point 11: In Figure 5, it must be included DTG to improve the discussion of the degradation events.

Response 11: DTG curve is included in Figure 5(b) as suggested

Point 12: Page 9, line 342: “The FTIR spectra of egg carton pulp and nitrocellulose are shown in Fig. 4.”. The authors must be substituted the number of Figure 4 for 6.

Response 12: The Figure 4 has been changed to Figure 6 as suggested

Point 13: FTIR spectra of egg carton pulp and nitrocellulose could be improved, including the main absorptions.

Response 13: The FTIR spectra of egg carton pulp and nitrocellulose has been improved as suggested

Point 14: The call of Table 4 must appear in the text.

Response 14: The call of Table 4 has included as suggested

Point 15: Page 10, line 382: The authors must be substituted the number of Table 3 for 5.

Response 15: The Table 3 has been changed to Table 5 as suggested

Point 16: In Table 5, acronyms B1, B2, B3, and B4 must be specified.

Response 16: The acronyms B1, B2, B3 and B4 has been specified as suggested

Reviewer 3 Report

The experimental article "Utilization of Recycled Egg Carton Pulp for Nitrocellulose as an Accelerant in Briquette Production" fully corresponds to the theme of the publication "Polymers". The research presented in the paper is fundamental and ambiguous. The article is written in clear, accessible language and will be of interest to a wide range of readers.
But it is necessary to substantially revise the article, namely:
1) There is a gross error in the first sentence (lines 30-31).
2) The introduction needs to be rewritten and structured. The literature that has already discussed cellulose nitrate as a gas pedal should be mentioned, and the first paragraph of the introduction should be rewritten to emphasize the use of cellulose nitrate as a gas pedal.
3) On line 42 there is a listing of non-traditional plant raw materials, and then the authors move on to egg cartons, but this type of raw material is not a plant raw material. It would be good to cite works on nitration of similar raw materials, maybe waste paper or similar.
4) Why didn't the authors determine the degree of polymerization of cellulose before nitration? It would be good to add this information.
5) Question on Table 1. What is the increase in lignin when treated with 1.5 M KOH compared to 1.0 M KOH?
6) Question on Table 1. What were the solubilities in ethanol, toluene, hot and cold water determined for? What do these values have to do with nitration?
7) Line 252: Hydroxyl? Could the hydrogen atom in the hydroxyl group be replaced?
8) The very idea of adding cellulose nitrates to briquettes is confusing. How stable would these briquettes be? The thermal stability of briquettes with cellulose nitrate added needs to be studied.

Author Response

Point 1: There is a gross error in the first sentence (lines 30-31).

Response 1: The (-NO3) has been changed to (-NO2)

Point 2: The introduction needs to be rewritten and structured. The literature that has already discussed cellulose nitrate as a gas pedal should be mentioned, and the first paragraph of the introduction should be rewritten to emphasize the use of cellulose nitrate as a gas pedal.

Response 2: The application of cellulose nitrate as a gas pedal has not been found in any literature. Back in 1980s, the cellulose nitrate are most famous used as a film base but cellulose nitrate film are unstable as seen in the yellowing and loss of flexibility. Due to its drawback, the cellulose nitrate were major shift to its solo use in explosives. Mostly the application of cellulose nitrate are based on its nitrogen content. The higher nitrogen content that is more than 12% typically been used as a main components of gun cottons, whilst the cellulose nitrate with low nitrogen content are used as wood coating, nail lacquer, automotive paints and leather finishers. This application of cellulose nitrate has been mentioned

Point 3:  On line 42 there is a listing of non-traditional plant raw materials, and then the authors move on to egg cartons, but this type of raw material is not a plant raw material. It would be good to cite works on nitration of similar raw materials, maybe waste paper or similar.

Response 3: The production of nitrocellulose mostly made from wood pulp. For example, tobacco stalks, Acacia mangium pulp, palm oil bunches (EFB) and bacterial cellulose. Apart from that, there is also some nitrocellulose made from non-plant raw materials such as cotton linters. Cotton linters is the main cellulose sources utilized in the manufacturing of nitrocellulose. This is because the most important part in the production of nitrocellulose is the cellulose contents itself. The other example of nitrocellulose sources has been revised and added

Point 4:  Why didn't the authors determine the degree of polymerization of cellulose before nitration? It would be good to add this information.

Response 4: The degree of polymerization has not been done in the meantime. In spite of that, the alpha-cellulose determination were based on TAPPI Standard T203 os-74.  The separation of the cellulose consists in pulps contains alpha-, beta- and gamma-cellulose. So, in my study I identify the alpha-cellulose because it contains higher-molecular weight of cellulose in pulp. In general, the alpha-cellulose indicates the pulp that resistant to the bleached process. The alpha-cellulose also supported by the delignification of pulp through the TAPPI Standard 222 om-02. This method has been mentioned

Point 5:  Question on Table 1. What is the increase in lignin when treated with 1.5 M KOH compared to 1.0 M KOH?

Response 5: The increment of lignin content when its reached at 1.5 M is might due to the bleaching process reached the final delignification phase. The reason of this increment happen has been included as suggested

Point 6:  Question on Table 1. What were the solubilities in ethanol, toluene, hot and cold water determined for? What do these values have to do with nitration?

Response 6: The ethanol-toluene, cold and hot water solubilities were done in order to determine the extractives content in pulp. The extractives are the minor components and a minor role in determining the pulp bleachability. Nevertheless, the extractives content can affect the bleachability of pulp negatively which may consuming more bleaching chemicals. So, this experiments were done to support more on lignin and also extractives. The role of this chemical constituents has been added as suggested

Point 7:  Line 252: Hydroxyl? Could the hydrogen atom in the hydroxyl group be replaced?

Response 7: The number of hydroxyl oxygen atoms affects the levels of nitration of nitrocellulose. This is because its affects both the number of donors and acceptors, so the number of bonds cannot be scaled proportionally to the number of -OH groups.

Point 8:  The very idea of adding cellulose nitrates to briquettes is confusing. How stable would these briquettes be? The thermal stability of briquettes with cellulose nitrate added needs to be studied.

Response 8: Based on the Td10% degradation shows the briquette with nitrocellulose degraded around 310℃ - 345℃, while the briquette without nitrocellulose degrade at 398℃. Based on TGA analysis of briquette, the nitrocellulose has enhanced the combustion of briquette. The thermogravimetric of briquette after additional of nitrocellulose has been added as suggested.

Round 2

Reviewer 2 Report

The manuscript has been sufficiently improved to warrant publication in Polymers.

Reviewer 3 Report

The articale is ready for publication